# Microstructures and Mechanical Properties of Deposited Fe-8Cr-3V-2Mo-2W on SCM420 Substrate Using Directed Energy Deposition and Effect of Post-Heat Treatment

**DOI:** 10.3390/ma14051231

**Published:** 2021-03-05

**Authors:** Ye Eun Jeong, Jun Yeop Lee, Eun Kyung Lee, Do Sik Shim

**Affiliations:** 1Department of Ocean Advanced Materials Convergence Engineering, Korea Maritime and Ocean University, Busan 49112, Korea; jye960302@naver.com; 2Interdisciplinary Major of Maritime AI Convergence, Department of Ocean Advanced Materials Convergence Engineering, Korea Maritime and Ocean University, Busan 49112, Korea; dhkekfdl2@naver.com (J.Y.L.); elee@kmou.ac.kr (E.K.L.)

**Keywords:** directed energy deposition (DED), tool steel, heat treatment, mechanical properties, carburization

## Abstract

In this study, the Fe-8Cr-3V-2Mo-2W tool steel powder was deposited on the SCM420 substrate through the directed energy deposition (DED) process. This study focuses on the mechanical properties of the deposited Fe-8Cr-3V-2Mo-2W and the effect of heat treatment on it. The changes in the microstructural characteristics of the deposited region due to heat treatment after deposition were observed. The influence of heat treatment on the mechanical properties was then analyzed accordingly and hence, the hardness, wear, impact and tensile tests were conducted on the deposited material. These properties were compared with those of the commercial tool steel powder M2-deposited material and the carburized specimen. In the deposited Fe-8Cr-3V-2Mo-2W layer, an increased martensite phase fraction was obtained through post-heat treatment and the amount of precipitated carbides was also increased. This increased the hardness from 48 to 62 HRc after heat treatment and the wear resistance was significantly improved as well. The amount of impact energy absorbed decreased from 11 J before heat treatment to 6 J after heat treatment, but the tensile strength significantly increased from 607 to 922 MPa. When compared with the M2-deposited surface, the Fe-8Cr-3V-2Mo-2W deposits had 3% lower surface hardness and 76% lower fracture toughness but exhibited 56% higher tensile strength. When compared with the carburized SCM420, the Fe-8Cr-3V-2Mo-2W deposits exhibited 3% higher surface hardness and wear resistance, 90% lower fracture toughness and 5% higher tensile strength. This study shows that surface hardening through DED can exhibit similar or superior mechanical properties when compared to carburizing.

## 1. Introduction

The mechanical components used in harsh environments can deteriorate due to wear and fatigue. In such cases, they can be repaired on-site, to reduce the cost of replacing them altogether. Therefore, in the manufacturing industry, improving the service life of parts (e.g., die/mold and mechanical parts) or rapidly repairing them at a low cost is important. Carburization is commonly used in mechanical parts and the carburizing process is typically performed either by allowing steel to absorb carbon atoms, or by diffusing the carbon atoms into a surface in a carbon atmosphere. The carbon content in the carburized layer hardened by quenching was increased to approximately 0.8 wt% [1]. This process can generate a hard surface that can withstand external wear on the surface. It is also commonly used for material hardening to withstand external impacts and loads [2]. According to a previous study [3], the application of the carburizing process to low-carbon steel with a hardness of 230 HV, can increase the hardness to 720 HV on a surface which has been sufficiently penetrated by carbon atoms. In the center, where the carbon penetration is insufficient, the hardness can be increased to 370 HV. In addition, the microstructure is improved by the high carbon content and a huge carbide network is formed [4,5]. The service life of components can be increased by increasing the penetration depth or the penetration time of carbon atoms during carburizing.

Various technologies [6] have been used as conventional methods for component repair, including gas tungsten arc welding [7], electronic beam welding and friction welding [7]. However, complications that arise during the welding process, such as pore generation, the heat-affected zone (HAZ) and thermal deformation, can affect the quality of products [8,9,10]. Laser cladding, which uses a high-power laser, can form a cladding layer with an exceptional bonding strength on the substrate surface by simultaneously melting the substrate and the metal wire. The surface characteristics of the substrate can be improved by cladding a high-hardness material on the substrate surface using this technology. In recent years, with the development of metal 3D printing technologies using a laser or an electron beam, technologies to reinforce the surface of metal components in various shapes have been applied. In particular, directed energy deposition (DED), an additive manufacturing (AM) technique (ISO/ASTM52900—15 Standard Terminology for Additive Manufacturing—General Principles—Terminology), which is used to manufacture various 3D products by depositing materials on a metal substrate using a layer-by-layer method, has been researched as a technology to repair metal components and reinforce surfaces [11]. This technology can produce parts with lower residual stress and better mechanical properties when compared to the parts produced by conventional methods [12,13,14]. It can also minimize the HAZ, which causes property degradation [15] and it has been commonly used for hardfacing by depositing high-hardness powder on the metal surface [16]. Owing to these benefits, the DED process is expected to replace the conventional methods.

This study was conducted on the deposition of high-hardness tool steel powder using DED for surface hardening. Various studies have been conducted on the deposition of tool steel powder using the DED process and on the analysis of the mechanical properties. Kim et al. [17] deposited the AISI M2 metal powder on the substrate surface using DED and improved the surface hardness by 30–50% when compared to that of the conventional tool steel. Lee et al. [18] evaluated the wear characteristics of specimens fabricated by AM, welded specimens and raw materials using the H13 hot work tool steel. The specimens fabricated by AM exhibited the highest hardness, whereas the welded specimens presented the lowest hardness. The specimens fabricated by AM also exhibited superior wear characteristics to those of the specimens produced by conventional welding. Powder metallurgical tool steels, such as CPM 10 V [19,20] and Vanadis 4 Extra [21], can create an extremely hard coating layer. The mechanical properties of such tool steels were also improved through heat treatment after deposition [22]. Junker et al. [23] performed heat treatment after deposition to reinforce the surface of the components to be used in high-temperature environments. The mechanical properties of the components were found to be superior to those of the raw material and demonstrated the potential of heat treatment to improve the service life of deposited tool steels. Leunda et al. [21] deposited CPM metal powder on a Vanadis4 substrate using DED and improved its wear resistance. Based on these preliminary studies, a study to increase the life of the actual blanking punch [24] and a study on the repair of the damaged components in a damaged mold made of H13 material were conducted [25]. According to Fukaura et al. [26], the hardness and the wear resistance in the deposited region improved in proportion to each other. It was established that this result was affected by the microstructure change and carbide dispersion in the deposited region. In addition, it was observed that high-speed tool steel (HSS) with high contents of tungsten, vanadium and molybdenum forms MC-based microcarbides, while melted and solidified by a laser, thereby improving the hardness and wear resistance characteristics of the surface [20].

However, metal powders with a high carbon content, such as HSS and CPM 15V metal powder, were observed to exhibit more interfacial cracks between the substrate and deposited region with an increase in the deposition area and the height of the hardened. These defects are induced due to the thermal stress caused by the difference in the thermal expansion coefficient as the rapid melting and solidification processes are repeated [27]. In welding, which has a mechanism similar to that of the laser AM process, substrate preheating, a method used to minimize the development of thermal stress by decreasing the cooling rate, is used to remove such defects. Such preheating, however, affects the hardness and the wear resistance as it inhibits the formation of hardened structures by decreasing the cooling rate of the weld [28]. Therefore, selecting an appropriate preheating temperature is required to form a perfect hardened layer without any defects. There is, however, no case in which substrate preheating was applied to the deposition process of alloy steel materials with a high carbon content, such as HSS and CPM 15 V and there has been no research on the microstructure and the wear resistance characteristics caused by preheating.

Thus far, most of the studies on surface hardening through metal deposition have focused on commercial tool steels. For example, commercial tool steel powder [17,20,21,29,30,31,32] was deposited on the die tool steel substrate [17,29,30,31] or a powder of the same material as the substrate [21,33,34] was deposited. However, studies conducted on new alloy powders are still insufficient. Therefore, this study attempts to reinforce the surface of a low-carbon steel substrate by depositing a newly combined alloy tool steel powder (Fe-8Cr-3V-2Mo-2W powder). The deposited specimens were subjected to heat treatment to obtain a further improved mechanical properties. These specimens were compared with a specimen surface-hardened by the carburizing process, which is a conventional hardening method. Additionally, M2, which is a commercial tool steel powder applied in surface hardening, was also included in the comparison. To compare the mechanical properties of these specimens, wear tests, impact tests and tensile tests were conducted. Through these tests, the improvement in the mechanical properties by surface hardening that used the new alloy powder, when compared to that of carburizing was examined. The mechanical properties were also analyzed in terms of the microstructural characteristics.

## 2. Materials and Methods

### 2.1. DED Process

Direct metal tooling (DMT) MX3 equipment, which is the DED machine developed by Insstek (Daejeon, South Korea), was used in this study. Figure 1 shows the schematic diagram of the process and a photograph of the equipment. This equipment consists of a 4 kW CO_2_ laser that serves as the heat source, a numerical control system, the MX-CAM software, a 5-axis NC machine tool, a powder feed system with three hoppers and a coaxial powder nozzle. The laser beam diameter was 1 mm. The principle of the DED process is similar to that of laser cladding. A melt pool is formed on the substrate surface by locally irradiating a high-power laser beam onto the substrate and metal powder (or wire) is simultaneously supplied to the melt pool in real time through the powder gas from the powder feed system for melting. The melt pool in which the substrate and the metal powder are mixed is rapidly cooled to form a deposited bead with a dense and fine crystalline structure. During the deposition process, argon gas was used to deliver the powder and also, to prevent the oxidation of the supplied powder. The deposited beads form a deposition layer along a specific path and this process is repeated and each layer is deposited in the vertical direction in a crisscrossing path, to form a three-dimensional structure. Table 1 shows the process parameters applied in the experiment.

### 2.2. Materials

In this study, SCM420, which is a chromium-molybdenum steel alloy, commonly used for mechanical components, was used as the substrate. The metal powder used for deposition was Fe-8Cr-3V-2Mo-2W which was supplied by the manufacturer (Rovalma Co., Barcelona, Spain). An AISI M2 HSS powder-deposited specimen was also fabricated for comparison. The M2 powder used (supplied by Carpenter Co., Incheon, South Korea), was spherical with a diameter ranging from 50 to 150 μm and contained chromium, molybdenum, tungsten and vanadium, which can form various types of carbides. Table 2 shows the chemical compositions of the substrate and Fe-8Cr-3V-2Mo-2W and M2 powders.

### 2.3. Specimen Preparation

To observe the changes in the properties due to heat treatment after the deposition of the Fe-8Cr-3V-2Mo-2W powder, heat treatment was performed through quenching and tempering. As shown in Figure 2, austenizing was performed at 840 °C for two hours and at 1060 °C for 40–45 min in a nitrogen purging chamber, followed by oil quenching at two bars. The tempering process of cooling in the air at 520 °C for two hours was then repeated thrice.

Hardness tests, wear tests, tensile tests and impact tests were conducted to evaluate the mechanical properties. The specimens deposited with the Fe-8Cr-3V-2Mo-2W powder (“as-built (FeCrVMoW)”), the specimen subjected to heat treatment after deposition (“as-built (FeCrVMoW)-PHT”), the specimen deposited with the M2 powder (“as-built (M2)”) and the carburized SCM420 (“C-SCM420”) were compared with each other.

In the carburizing process of SCM420, primary carburizing was performed by maintaining the substrate at 1193 K for 36 min in a 1% carbon atmosphere after degreasing. Secondary carburizing was performed by maintaining the substrate at 1133 K for 22 min and oil quenching was then performed. Finally, tempering was performed at 433 K for 17 min.

### 2.4. Observation of Microstructures

The sample was cut to observe the microstructure, ground up to #200–#2400 and polished up to 3 µm and 1 µm in a sequential manner. An etching solution (HNO3: 79.36 mL and distilled water: 20.64 mL) was used at an applied voltage and current of 2 V and 2 A, respectively. The microstructure of the etched surface was examined using a field emission scanning electron microscope (FE-SEM, MIRA3, TESCAN, Seoul, South Korea).

To investigate the crystal structure, X-ray diffraction (XRD, Smartlab, 9 kW, Rigaku, Tokyo, Japan) was employed using Cu Ka radiation (λ = 1.54 Å) at 40 kV and 30 mA. The beam size and step size were 2.0 mm and 0.01, respectively. Separate X-ray measurements of the deposited layers and the substrate were recorded to prevent peaks from each region from overlapping.

### 2.5. Mechanical Testing

The hardness was measured by observing the indented area while applying a load of 0.1 kgf for ten seconds using a micro-Vickers hardness tester (Akashi, HM-122, Sakado, Japan). The Vickers hardness was measured at 30 points dispersed along 0.1 mm intervals in the vertical direction from the deposited region to the substrate and the Rockwell hardness was measured at five points dispersed along 0.1 mm intervals in the horizontal direction on the surface of the deposited region.

The specimen for the wear test was fabricated with a 10 mm height, including the 1 mm deposited region on top of the cylindrical substrate with a 30 mm diameter and a 9 mm height, as shown in Figure 3a. Prior to the wear test, the surfaces of all the specimens were polished and cleaned with ethanol. The test was conducted for an hour at a rate of 60 rpm using an alumina (Al_2_O_3_) ball (7 mm diameter) with a load of 10 kgf with the help of a ball-on-disc tester. The lost weight and the worn depth were measured to compare the wear resistance and the wear track was observed using a field emission scanning electron microscope (FE-SEM).

The Charpy impact test was conducted to evaluate the fracture toughness and the interfacial bonding between the deposited region and the substrate. As shown in Figure 3b, the specimens were fabricated to include the 5 mm deposited region. The test was repeated thrice under each condition and the average values were extracted. It was conducted at a temperature of 22 ± 1 °C and a humidity of 43 ± 3%, in accordance with the ASTM A370 standard. After the test, the impact absorbed energy was calculated and the fracture surface was observed using SEM.

For the tensile test, specimens with the 1.5 mm deposited region and the 1.5 mm substrate were fabricated in accordance with ASTM E8, as shown in Figure 3c. The test was conducted at a rate of 1 mm/min using a 50 kN tensile tester (MTS, 322.21, Eden Prairie, MN, United States). After repeating the test thrice, the strength and the elongation were extracted through the average values. After the test, the fracture surface was observed using the FE-SEM.

## 3. Results and Discussion

### 3.1. X-ray Diffraction

Figure 4 shows the morphology of the specimen fabricated by depositing the Fe-8Cr-3V-2Mo-2W powder on the SCM420 substrate. The SEM image of the interface shows that there are no cracks or pores present in the deposited region and near the interface between the deposited region and the substrate. XRD analysis was conducted to examine the phase, the carbide and the microstructure formed in the deposited zone. Figure 5 shows the XRD patterns of the C-SCM420, the as-built (M2), the as-built (FeCrVMoW) and the as-built (FeCrVMoW)-PHT, respectively. Figure 5a shows the ferrite (α), the martensite (α’) and the austenite (γ) peaks, but there was no carbide detected. Owing to the high carbon content of the carburized layer, the retained austenite transforms into stable martensite when cooled to room temperature. In addition, a high martensite peak was detected because most of the austenite transformed into martensite during the oil quenching after the carburizing [35]. In contrast, to the other specimens that did not exhibit a carbide peak, the peak of tungsten carbide was observed in the austenite peak in the as-built (M2)(Figure 5b). According to C. Navas [32], the type of carbide detected in Figure 5b is considered the M_6_C-type Fe_3_W_3_C carbide. The M6C carbide can become a nucleus for the crystallization of the austenite because it is not completely dissolved during the deposition process [36]. This allows for the tungsten carbide detected through the XRD pattern in Figure 5b, to have the same peak as that of austenite. In the as-built (FeCrVMoW) and the as-built (FeCrVMoW)-PHT (Figure 5c,d), only the austenite and martensite peaks were detected. The austenite and martensite structures are formed by the melting and the quenching of the metal powder as well as the high temperature gradient with the substrate. The XRD patterns in Figure 5c,d show that the microstructure was changed by heat treatment after deposition. During the quenching to room temperature in the austenite state, there is not enough time for the diffusion of carbon. Therefore, carbon remains in the austenite. However, through three tempering processes, the retained austenite transforms into martensite and the martensite into tempered martensite. This showed a decrease in the austenite fraction and an increase in the martensite fraction in the matrix.

### 3.2. Microstructure

Figure 6 shows the microstructure of the cross-section of each specimen. Figure 6a shows the cross section of the carburized substrate, which is mainly divided into three zones: the carburized layer with the highest carbon content (1), the diffusion zone between the carburized layer and the substrate (2) and the substrate zone with the lowest carbon content (3). In zone (1), a high hardness can be expected. Zones (1) and (2) are difficult to distinguish based on the microstructure, but they can be divided through the hardness of each zone. Figure 6b shows the plate martensite, which is a typical high-carbon martensite, along with some retained austenite. Because the carburized zone has a high carbon content, the microstructure of the C-SCM420 always includes the retained austenite [37].

Figure 6c shows the M2 deposited zone and interface of the as-built (M2). The interface of the bead exhibits a high cooling rate owing to the high thermal gradient between the substrate and the deposited zone. Therefore, columnar dendrites were formed in the direction of the heat transfer. Equiaxed grains were then formed in the center of the bead, where the cooling was relatively delayed. In the deposited region (Figure 6d), equiaxed grains were observed. A microstructure in which martensite is surrounded by an austenite network was observed and rod-like, as well as spherical carbides, were precipitated at the grain boundaries. Figure 6e shows the deposited region and interface of the as-built (FeCrVMoW). Similarly with the as-built (M2) as shown in Figure 6c, columnar and equiaxed grains were formed at the interface. The grain size in the deposited zone (Figure 6f) is larger than that in Figure 6d and small spherical and rod-like carbides were observed. The interface of the as-built (FeCrVMoW)-PHT shows similar microstructures (Figure 6g). However, the carbide precipitated in the as-built (FeCrVMoW)-PHT (as shown in Figure 6h) has a different shape from that of the one in the as-built (FeCrVMoW). Spherical and rod-like carbides were hardly observed, but agglomerated eutectic carbides were precipitated at the grain boundaries. This indicates that a phase transformation has taken place and a new carbide phase was generated during the heat treatment.

Figure 7 shows the EDS observation results for the carbides precipitated in the deposited region of the as-built specimens. Figure 7a shows the carbides of the as-built (M2). The precipitated carbides were composed of chromium, molybdenum, tungsten and vanadium, which are the main elements of carbides that can be generated in tool steel powder. The tungsten content was especially high, which can also be confirmed by the XRD pattern in Figure 5b. According to Kim et al. [38], the carbides in the deposited region of the as-built (M2) are inferred to be W-rich M_6_C carbides. As for the carbides in the as-built (FeCrVMoW), Cr-rich carbides were precipitated in contrast, to the carbides shown in the as-built (M2). Leunda et al. [39], has stated that the rod-like carbides are Cr-rich carbides. In the as-built (FeCrVMoW)-PHT, the agglomerated eutectic carbides are also Cr-rich carbides. Eutectic carbides that are precipitated in Cr-rich materials are M_7_C_3_ carbides [40,41]. As chromium carbides precipitated at grain boundaries are agglomerated in a narrow zone as eutectic carbides during the heat treatment process, they exhibit a higher chromium content than the carbides in Figure 7b.

### 3.3. Microhardness

Figure 8a shows the micro-hardness values measured along the depth direction from the carburized layer or the deposited region, to the substrate. In the C-SCM420 specimen, as the carbon content decreased from the top surface to the center, the hardness simultaneously decreased towards the center. The specimens fabricated by DED exhibited higher hardness in the deposited region than in the substrate regardless of the powder or heat treatment and the hardness sharply decreased at the interface. The slight difference in the hardness values at each position can be attributed to irregularity in the powder supply and fluctuations in the laser power. In addition, the irregular distribution of carbides in the deposited zone may also cause different hardness values at each position. The decrease in the hardness at the interface is due to the formation of a mixed layer caused by the simultaneous melting of the substrate and the powder. As the substrate has a relatively lower hardness than the deposited powder, the hardness decreases, depending on the mixing ratio of the substrate and deposited powder. The deposited region of the as-built (M2) exhibited a high hardness of 780 HV even though heat treatment was not applied. This could be due to the presence of elements such as V and W, contained in M2, as well as the presence of martensite and the precipitated carbides. The average hardness of the deposited region of the as-built (FeCrVMoW) was 672 HV, but it increased to 780 HV through post heat treatment. Previous studies on the deposition of tool steel [20,39] reported that secondary hardening occurred because the retained austenite transformed into tempered martensite along with the precipitation of secondary carbides. Therefore, the martensite fraction and the concentration of chromium carbides increased by heat treatment after deposition, which were confirmed through XRD analysis, microstructure observation and EDS results, are the causes of the increase in the hardness. Figure 8b shows the Rockwell hardness distribution on the surface of the deposited region. The surface hardness was 60 HRc for carburized C-SCM420 and 64 HRc for the as-built (M2). The as-built (FeCrVMoW) had the lowest surface hardness of 48 HRc, but this increased by 29% to 62 HRc after heat treatment due to the increased martensite fraction. Although the as-built (FeCrVMoW)-PHT had a lower martensite fraction and a smaller amount of carbide alloy elements than the as-built (M2), its hardness was similar to that of the as-built (M2).

### 3.4. Wear Test

The wear test was conducted to compare the wear resistance under each condition. After the test, the weight of each specimen before and after the test, was compared and the wear rate was examined by observing the width of the wear track generated on the surface. In all the specimens, the wear tracks were generated by the friction between the ball and the surface of each wear specimen, as shown in Figure 9. The wear track of C-SCM420 formed a wear width of 814 μm and that of the as-built (M2) formed the narrowest width of 660 μm. The as-built (FeCrVMoW) formed the widest wear track of 1030 μm, but this decreased to 798 μm after the heat treatment. In other words, the width of the wear track decreased as the surface hardness increased.

Figure 10 compares the weight loss of each specimen. The weight loss of C-SCM420 was 5.0 mg, which has an even high surface hardness. The as-built (M2) with the highest surface hardness exhibited the lowest weight loss of 0.3 mg. The weight loss of the as-built (FeCrVMoW) with the lowest surface hardness was 6.33 mg, indicating that it is the most vulnerable to wear. The weight loss of the heat-treated specimen, however, decreased to 2.32 mg. Here, the wear track width and weight loss of the C-SCM420 are notable. The C-SCM420 exhibited a small wear track width due to its high surface hardness, but it showed a relatively large weight loss. This indicates that once the thin carburized layer on the surface is abraded by the ball, the wear rapidly proceeds in the substrate which is less carburized. Conversely, in the deposited specimens, no remarkable loss was observed because of the thick hardened layer (1 mm). These results show that the hardfacing through DED is superior to the conventional carburizing method in terms of wear resistance.

Figure 11 shows the SEM images and the 3D topography of the worn surfaces. The alumina ball used in the wear test begins to slide and applies a load to the surface of each specimen. During the test, elastoplastic deformation occurs at the positions where there is friction, resulting in peeling, grooves and debris. The wear surfaces of the as-built (FeCrVMoW) and the C-SCM420, which exhibited high weight losses after the wear test, were much rougher than those of the as-built (FeCrVMoW)-PHT and the as-built (M2). The C-SCM420 (Figure 11a) shows a combination of abrasive wear and adhesive wear. In addition, micro-carbides were released due to a large and a long load on the surface, resulting in abrasive debris. The released microcarbides generated deep grooves on the surface of the specimen, thereby causing a large loss of weight, due to which, the plowing groove (PG) caused by the tearing and the peeling of the material on the surface were also observed. The wear debris is adhered along the sliding direction on the surface of C-SCM420 due to severe wear. The adhered debris is irregularly distributed in the lowest zone (blue region) in Figure 11b. In contrast, the as-built (M2) specimen shows peeling, slight spalling and microgrooves and the deepest wear track (blue zone in Figure 11d), exhibits the narrowest width. The W-rich and Mo-rich carbides are harder and more resistant to wear than Cr-rich carbides [42]. In addition, the wear resistance of the tool steel containing a high proportion of alloying elements and complex carbides is highly dependent on the composition and the types of the carbides [20]. Therefore, it can be inferred that the as-built (M2), containing carbides mixed with a high composition of alloying elements, has the highest wear resistance. The as-built (FeCrVMoW) (Figure 11e) shows wide and deep grooves. Additionally, some debris was observed to be adhered to the surface and the spalling. The rod-like carbides which appear to be precipitated at the grain boundaries of the as-built (FeCrVMoW), were released and they deeply eroded the material surface under the continuous load, resulting in spalling and the generated debris was adhered onto the track. Conversely, the as-built (FeCrVMoW)-PHT (Figure 11g), shows spalling at a shallow depth across the image and the generated debris is adhered to the edge of the wear track. It can be observed in Figure 6h, that the surface was not smoothly worn due to the unevenly distributed eutectic carbides. In addition, wedges, which are plastic flows, were observed along with microgrooves. As the abrasive debris generated around the wedges continuously scratch the surface and are pushed out of the matrix, they can cause significant loss of material during the wear test. However, there was no severe wear as the phase transformation of martensite improved the wear resistance [43]. In addition, as the carbides in the microstructure were evenly and widely distributed, they appear to have effectively protected the matrix from repeated friction, thereby increasing the wear resistance.

### 3.5. Impact Test

The Charpy impact test was conducted to evaluate the fracture toughness of the material. The impact applied to each specimen was allowed to propagate to the substrate by causing a crack in the notch. Figure 12 shows the fractured surfaces of the specimens after the impact test. The crack that initiated at the notch did not propagate along the interface between the deposited layer and the substrate (Figure 12b–d). This indicates that the interface had excellent bonding strength. In C-SCM420, cracks occurred at the interface between the carburized layer and the substrate. The impact energy led to crack propagation towards the interface of the carburized layer. A large amount of impact energy can be absorbed by the substrate, which is relatively ductile when compared to the deposited region or the carburized layer. This can be confirmed by the cross-section (area reduction) of the plastically deformed specimen after the impact test. C-SCM420, which had a larger proportion of the ductile substrate than the deposited specimens, exhibited the largest plastic deformation (largest area reduction). This indicates that the impact energy was absorbed in the form of plastic deformation energy before the final fracture of the specimen. Conversely, the plastic deformation was not large for the deposited specimens. However, among these specimens, the as-built (M2) (Figure 12b) exhibited a larger plastic deformation than the as-built (FeCrVMoW) (Figure 12c) and the as-built (FeCrVMoW)-PHT (Figure 12d). Figure 12d indicates the smallest plastic deformation in the substrate. Significant plastic deformation was not observed due to the increased brittleness of the substrate and the deposited region, caused by heat treatment after deposition. Therefore, a decrease in the fracture toughness, can be expected after heat treatment.

Figure 13 shows the absorbed impact energy for each specimen. C-SCM420, which contained the largest region of the un-carburized substrate, exhibited the highest impact absorption energy, of 60.5 J. When an impact load is applied to the C-SCM420, the small carbides and the fine grains can absorb high impact energy as they increase the complexity of the crack propagation path. In the case of the as-built (M2), Figure 6d shows the presence of small grains and carbides. Therefore, it exhibited a high impact energy absorption of 25 J. Conversely, the as-built (FeCrVMoW), with large grains and carbides exhibited a low impact energy absorption of 11 J. Its impact energy absorption decreased by 45% to 6 J after heat treatment. The toughness of the as-built (FeCrVMoW)-PHT decreases due to the increased fraction of the martensite and the eutectic carbides precipitated at the grain boundaries. Particularly, the carbides precipitated at the grain boundaries cause stress concentration, thereby inducing the easy propagation of cracks at the grain boundaries. Therefore, the as-built (FeCrVMoW)-PHT has a lower impact energy absorption than the as-built (FeCrVMoW).

Figure 14 shows the SEM images of the fractured surface. The fractured zone just below the U-notch of C-SCM420 (Figure 14a) was composed of numerous dimples which indicate a ductile fracture. In contrast, since the cracks generated around the U-notch rapidly propagated along the downward direction, a transgranular fracture, a type of brittle fracture, was observed in the center region of C-SCM420 (Figure 14b). The deposited specimens also exhibited similar topograph, with slight differences arising from the deposited material. The deposited regions mainly exhibited the geometry of the intergranular fracture. The as-built (M2) shows the geometry of a dense intergranular fracture. The complex fractured surface significantly contributes to the high toughness due to the fine grains and this is one of the mechanisms that can improve the fracture toughness of tool steel [29]. In addition, unlike the other specimens, the agglomerated dimples shown in Figure 14d, are distributed over a large area. Therefore, the specimen had a high toughness because of a ductile fracture caused by the dimples, unlike the substrate of other specimens which exhibited a brittle fracture. For the as-built (FeCrVMoW) (Figure 14e), dendrites with multiple arms were observed along with an intergranular fracture. Microcracks present in some areas reduce the toughness because they lead to a stress concentration. However, the as-built (FeCrVMoW) has a higher toughness than the as-built (FeCrVMoW)-PHT owing to the high fraction of the austenitic matrix. At the interface of the as-built (FeCrVMoW) (Figure 14f), a transgranular fracture and a local area with concentrated microdimples, were observed. The deposited regions of the as-built (FeCrVMoW) and the as-built (FeCrVMoW)-PHT, have similar fracture surfaces, regardless of heat treatment. Overall, intergranular fractures and transgranular fractures were both observed, with intergranular fractures occurring more frequently. In addition, the deposited region of the as-built (FeCrVMoW)-PHT (Figure 14g), exhibited a greater complexity in the fractures. Brittle fractures and dimples agglomerated in a small area were also observed in the substrate.

### 3.6. Tensile Test

Figure 15 shows the tensile test results. C-SCM420 exhibited high tensile strength (877 MPa), owing to the high martensite fraction present in the carburized zone of the surface. It also showed the highest elongation (9.5%) due to the un-carburized substrate. The as-built (M2) exhibited a tensile strength of 592 MPa and an elongation of 6.3%. It had the lowest tensile strength among the deposited specimens but exhibited a high elongation value. As confirmed by the results of the impact test, it exhibited high elongation under a static load along with a high impact toughness. The as-built (M2) appears to have low tensile strength since small carbides cannot be effectively bonded and may cause stress concentration. The cracks generated in the deposited region, which is more brittle than the substrate, propagated to the substrate, thereby causing the final fracture. The tensile strength of the as-built (FeCrVMoW) was 607 MPa and its elongation was very low, at less than 1%. After the heat treatment, the tensile strength was increased by approximately 52% to 922 MPa, but the elongation decreased.

Figure 16 shows the low-magnification images of the cross section of the fractured specimens. As the carburized zone of C-SCM420 (Figure 16a) was highly brittle, its fractured surface was flat. The specimen, however, had high elongation due to the plastic deformation of the un-carburized substrate. The as-built (M2) (Figure 16b) exhibited necking in the substrate zone, unlike the as-built (FeCrVMoW) (Figure 16c) and the as-built (FeCrVMoW)-PHT (Figure 16d) and defects such as pores and cracks were not observed. The elongation in the substrate was greater when compared to that of the deposited region. In the case of the specimens deposited with FeCrVMoW (Figure 16c,d), the fracture is observed to be caused due to the stress concentration caused by the micropores present in the deposited region. It is estimated that the initial crack originated from the pores present in the deposited region and rapidly propagated to the substrate. Unlike the C-SCM420 and the as-built (M2), no plastic deformation was observed in the substrate or in the deposited region. This indicates that the pores present in the deposited region led to rapid fracture without a distinct elongation for the as-built (FeCrVMoW) and the as-built (FeCrVMoW)-PHT specimens.

Figure 17 shows the SEM images of the fracture surfaces. In the carburized zone of C-SCM420 (Figure 17a), brittle fractures was observed along with some microdimples, which can be attributed to the high carbon content in the carburized zone. In the non-carburized layer (Figure 17b), however, dimples were observed, indicating ductile fracture. Essentially, for the C-SCM420 specimen, the carburized zone led to the high strength and the un-carburized substrate led to the high elongation. The deposited region of the as-built (M2) (Figure 17c) exhibited a slightly coarser intergranular fracture than that of the as-built (FeCrVMoW)-PHT (Figure 17g) and a finer intergranular fracture surface than that of the as-built (FeCrVMoW) (Figure 17e). The tensile strength was also higher in the same order. The fracture surface of the substrate of the as-built (M2) (Figure 17d), however, exhibited different topograph from the substrate of the as-built (FeCrVMoW) (Figure 17f) and that of the as-built (FeCrVMoW)-PHT (Figure 17h), both of which, exhibited brittle fractures. In Figure 17d, a ductile fracture was observed due to the presence of shallow dimples. In other words, while the substrate was elongated, the specimen was fractured due to the crack propagated from the deposited region. The deposited region of the as-built (FeCrVMoW) (Figure 17e) exhibited an intergranular fracture with large grains as also shown in the fractured surface of the impact specimen of the as-built (FeCrVMoW) (Figure 14e). The substrate of the as-built (FeCrVMoW) (Figure 17f) showed a transgranular fracture, which is a rapid brittle fracture, with river patterns. The deposited region of the as-built (FeCrVMoW)-PHT (Figure 17g) exhibited the finest intergranular fracture and its substrate (Figure 17h) showed a quasi-cleavage fracture and river patterns. Its strength was greater when compared to that of the as-built (FeCrVMoW), due to the fine grains.

## 4. Conclusions

In this study, the Fe-8Cr-3V-2Mo-2W powder was deposited on the SCM420 substrate using a laser and post-deposition heat treatment was performed. The microstructure and the XRD pattern of Fe-8Cr-3V-2Mo-2W subjected to post heat treatment, were analyzed and mechanical tests were conducted. The results were compared with those of the carburized specimen and the specimen deposited with commercial tool steel M2.
(1)The microstructure of the deposited Fe-8Cr-3V-2Mo-2W layer formed fine equiaxed grains because the melt pool was formed and cooled rapidly. During the heat treatment, a phase transformation occurred in the Fe-8Cr-3V-2Mo-2W layer, thereby decreasing the austenite fraction and increasing the martensite fraction.(2)The high hardness of the as-built (FeCrVMoW)-PHT also affected its wear resistance. The wear track of the as-built (FeCrVMoW)-PHT exhibited a lower value (798 μm) when compared to those of the as-built (FeCrVMoW) (1030 μm) and C-SCM420 (814 μm). This was due to the influence of the MC-type carbides present in the as-built (FeCrVMoW)-PHT and the generated carbides effectively protected the matrix.(3)The as-built (FeCrVMoW)-PHT exhibited a lower toughness value than the as-built (FeCrVMoW) because its brittleness increased due to the increased martensite after heat treatment. The as-built (FeCrVMoW) had a higher impact toughness because its small carbides made the crack propagation more complicated.(4)The tensile strength increased from 607 to 922 MPa after the post-deposition heat treatment. This is due to the finer grains as well as the decrease in the retained austenite and the increase in the martensite fraction described above.

In this study, it was confirmed that surface hardening through DED can exhibit similar or superior mechanical properties (excluding fracture toughness) when compared to carburizing, a conventional method for engineering steel. However, the low elongation of the deposited material is yet to be solved. Consequently, further research is required on heat treatment which can improve the mechanical properties of the deposited region and the substrate simultaneously. In addition, the optimization of the parameter conditions to reduce defects (pores in this study) that may occur in the Fe-8Cr-3V-2Mo-2W deposition process further hinders the prevention of premature fracture.

## Figures and Tables

**Figure 1 materials-14-01231-f001:**
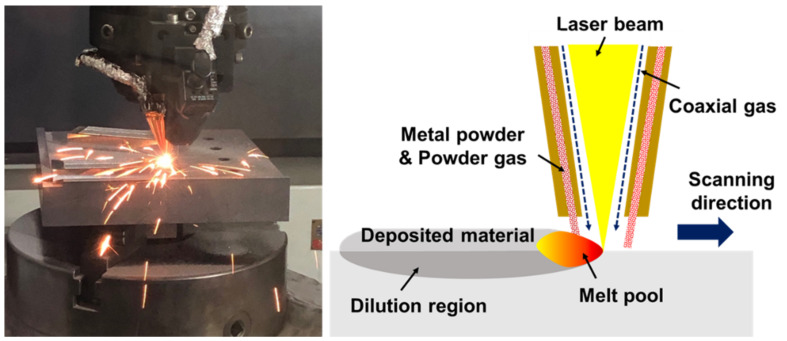
Schematic diagram of the directed energy deposition (DED) process.

**Figure 2 materials-14-01231-f002:**
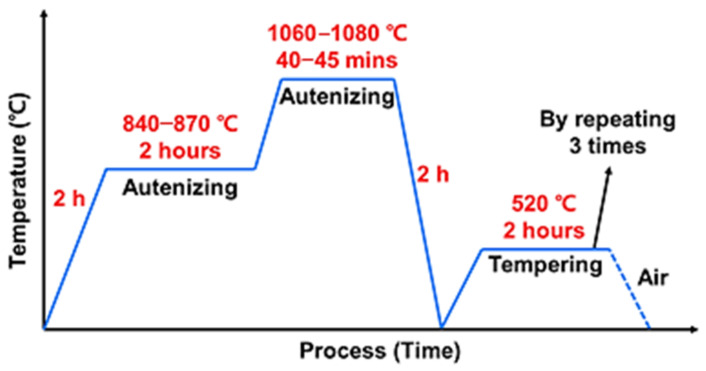
Schematic representation of the heat-treatment process.

**Figure 3 materials-14-01231-f003:**
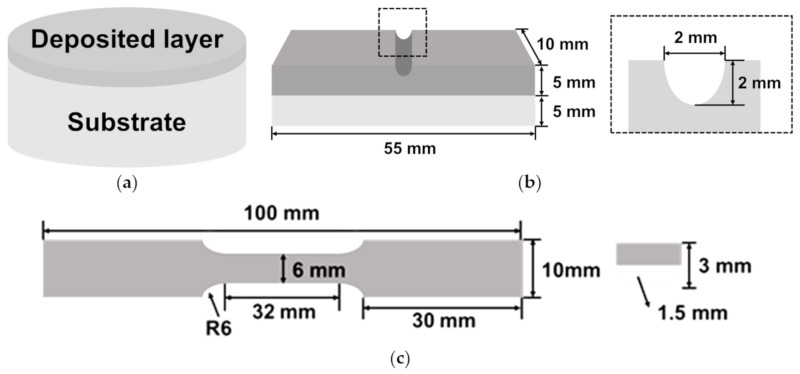
Schematic representation of the specimens for the (**a**) wear test, (**b**) U-notch Charpy impact test and (**c**) tensile test.

**Figure 4 materials-14-01231-f004:**
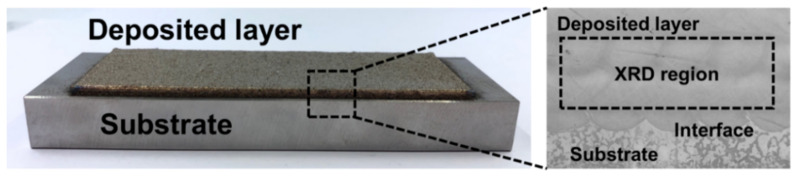
Cross section of deposited specimen for XRD analysis.

**Figure 5 materials-14-01231-f005:**
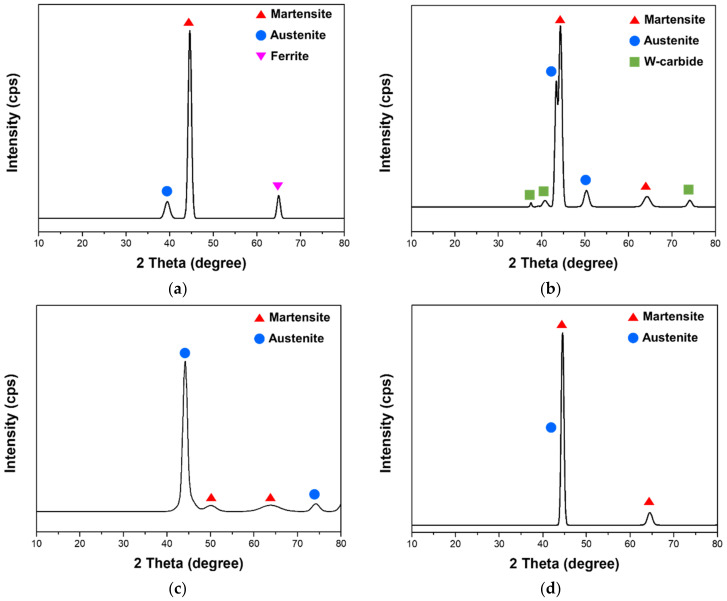
X-ray diffraction patterns of (**a**) C-SCM420, (**b**) as-built (M2), (**c**) as-built (FeCrVMoW) and (**d**) as-built (FeCrVMoW)-PHT.

**Figure 6 materials-14-01231-f006:**
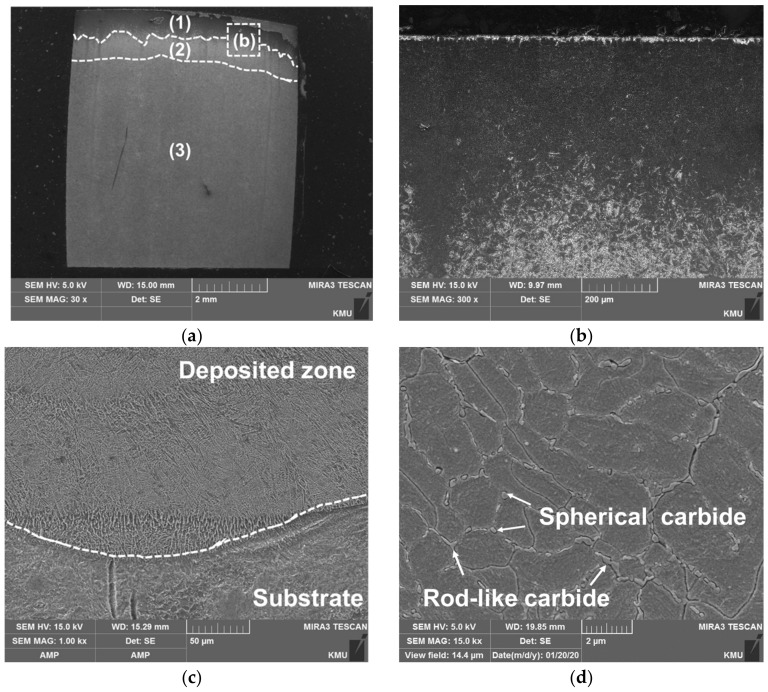
SEM micrographs of the microstructures: (**a**) C-SCM420, (**b**) carburized zone, (**c**) interface be-tween M2 deposits and substrate, (**d**) M2 deposited region, (**e**) interface between FeCrVMoW deposits and substrate, (**f**) FeCrVMoW deposited region, (**g**) interface region of as-built (FeCrVMoW)-PHT and (**h**) deposited layer of as-built (FeCrVMoW)-PHT.

**Figure 7 materials-14-01231-f007:**
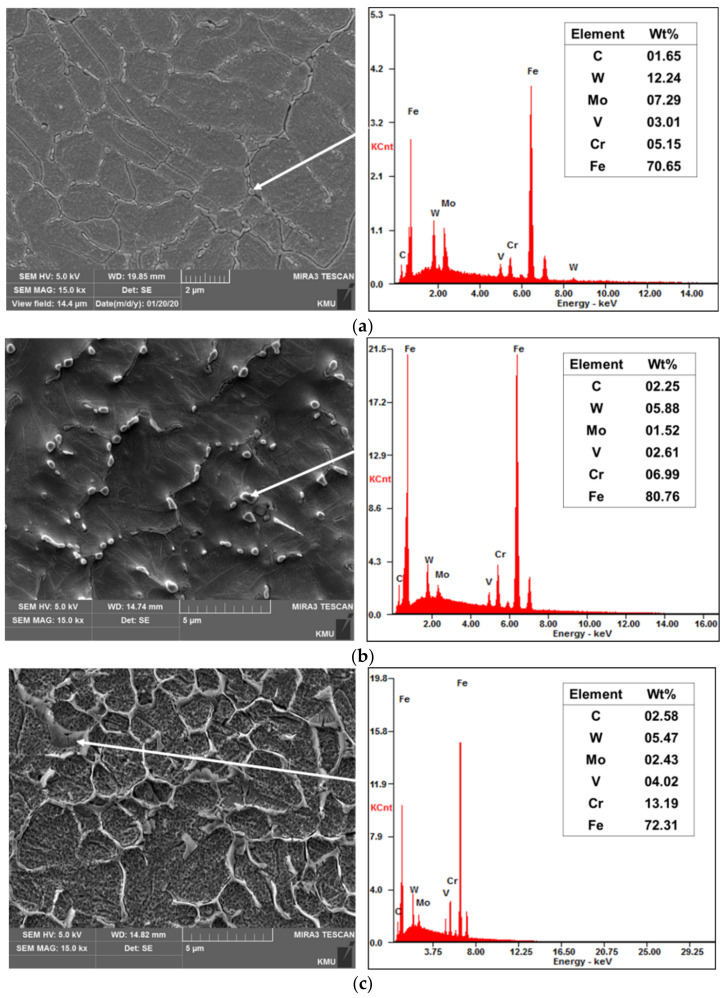
Results of the EDS composition analysis: (**a**) as-built (M2), (**b**) as-built (FeCrVMoW) and (**c**) as-built (FeCrVMoW)-PHT.

**Figure 8 materials-14-01231-f008:**
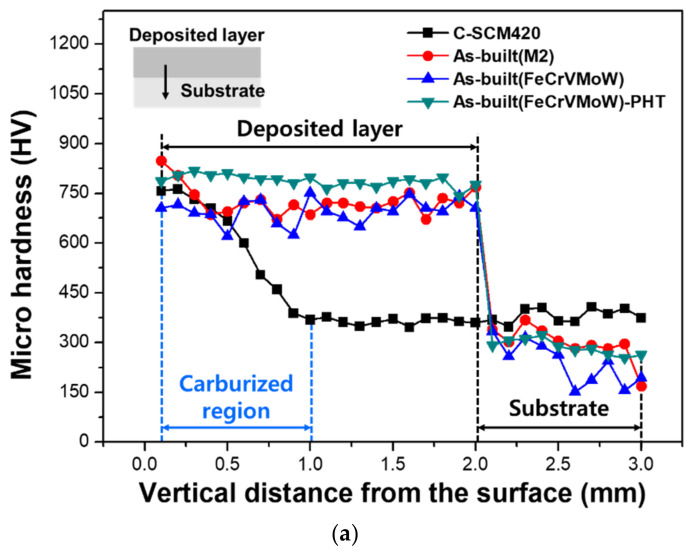
Hardness test profile in the (**a**) vertical direction and (**b**) horizontal direction.

**Figure 9 materials-14-01231-f009:**
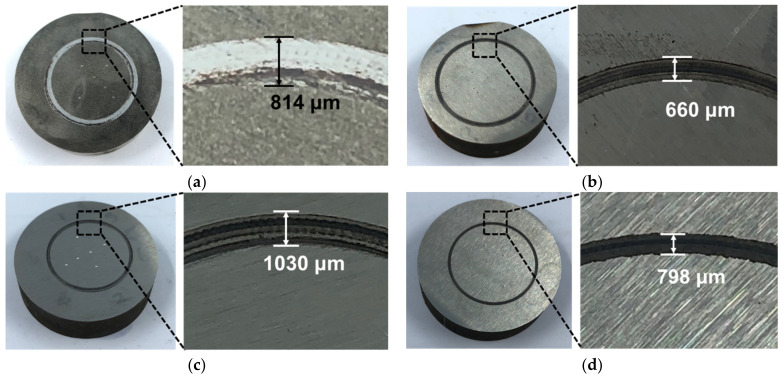
Images of wear scars after ball-on-disc wear test; (**a**) C-SCM420, (**b**) as-built (M2), (**c**) as-built (FeCrVMoW) and (**d**) as-built (FeCrVMoW)-PHT.

**Figure 10 materials-14-01231-f010:**
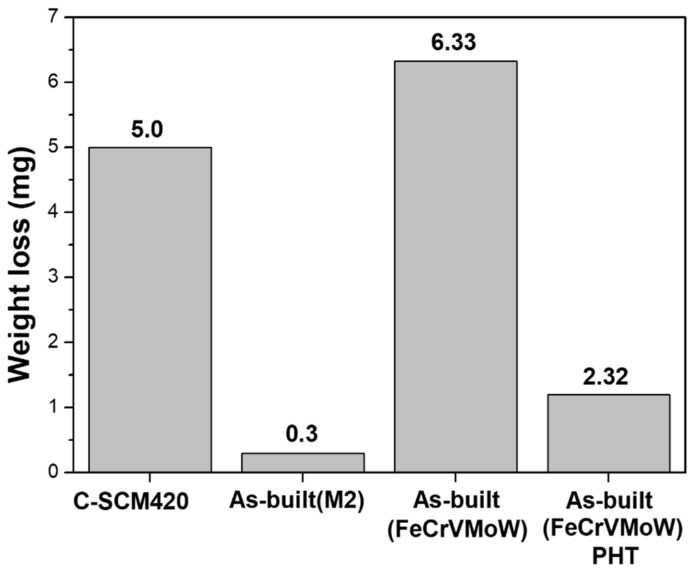
Comparison of weight loss of each specimen.

**Figure 11 materials-14-01231-f011:**
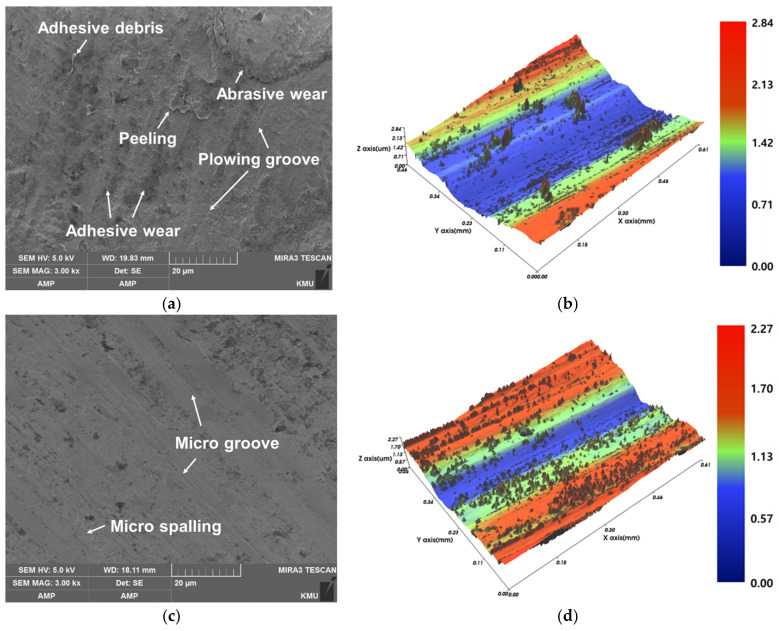
SEM micrographs and 3D topographies of the worn surface: (**a**,**b**) C-SCM420, (**c**,**d**) as-built (M2), (**e**,**f**) as-built (FeCrVMoW), (**g**,**h**) as-built (FeCrVMoW)-PHT.

**Figure 12 materials-14-01231-f012:**
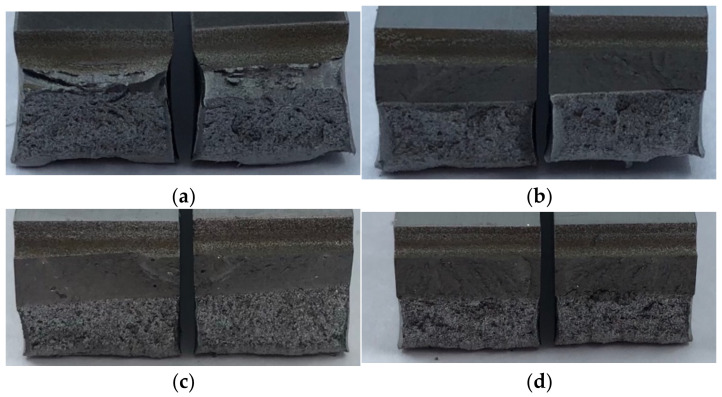
Fracture specimen after the Charpy test: (**a**) C-SCM420, (**b**) as-built (M2), (**c**) as-built (FeCrVMoW) and (**d**) as-built (FeCrVMoW)-PHT.

**Figure 13 materials-14-01231-f013:**
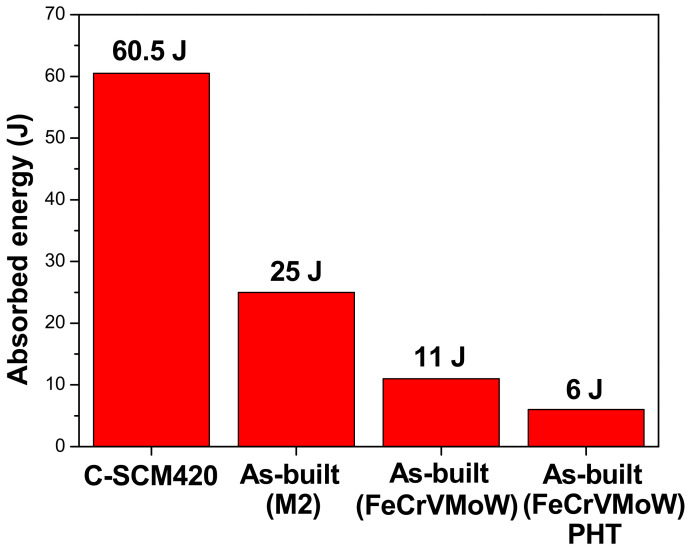
Comparison of average absorbed energies after the Charpy impact test.

**Figure 14 materials-14-01231-f014:**
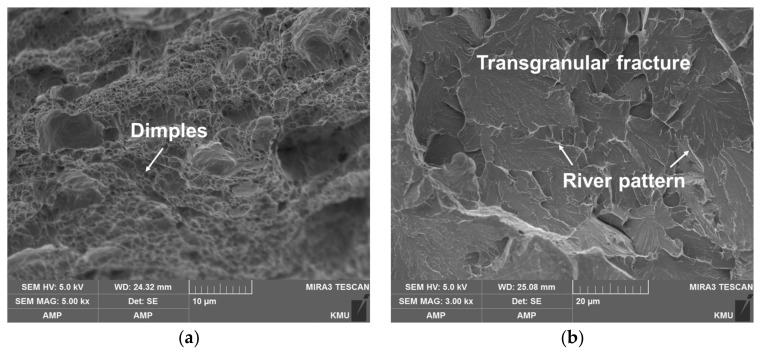
Micro-images of the fracture surfaces: (**a**) carburized layer of SCM420, (**b**) center region of C-SCM420, (**c**) deposited layer of as-built (M2), (**d**) interface of as-built (M2), (**e**) deposited layer of as-built (FeCrVMoW), (**f**) interface of as-built (FeCrVMoW), (**g**) deposited layer of as-built (FeCrVMoW)-PHT and (**h**) interface of as-built (FeCrVMoW)-PHT.

**Figure 15 materials-14-01231-f015:**
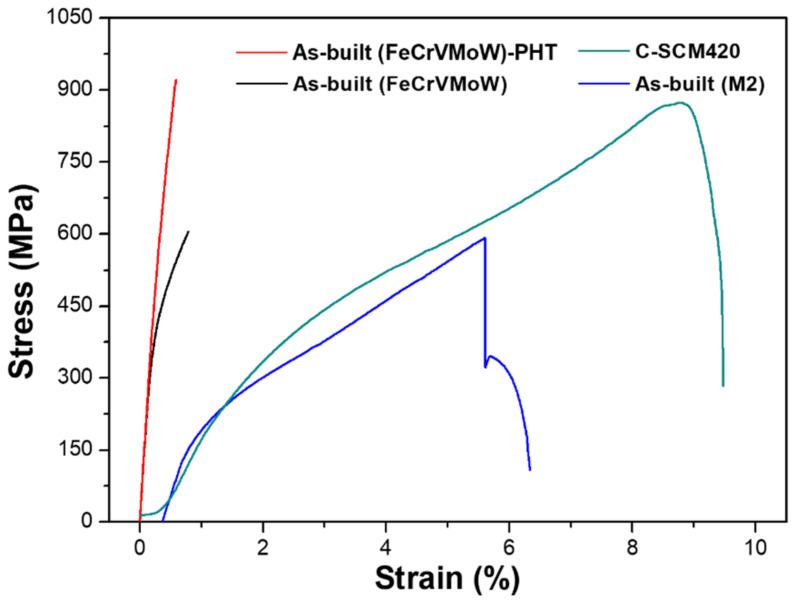
Engineering stress-strain curve of each sample.

**Figure 16 materials-14-01231-f016:**
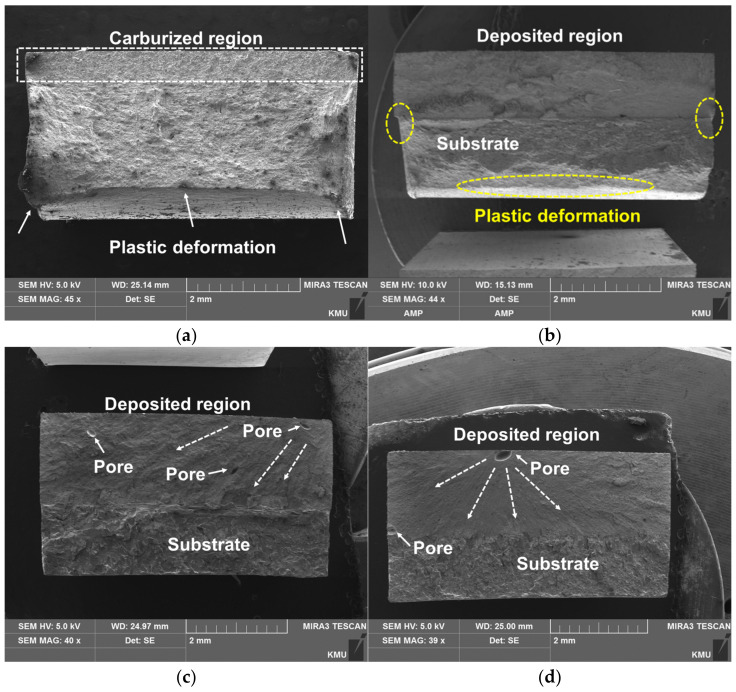
Fracture surfaces of the tensile specimens: (**a**) C-SCM420, (**b**) as-built (M2), (**c**) as-built (FeCrVMoW) and (**d**) as-built (FeCrVMoW)-PHT.

**Figure 17 materials-14-01231-f017:**
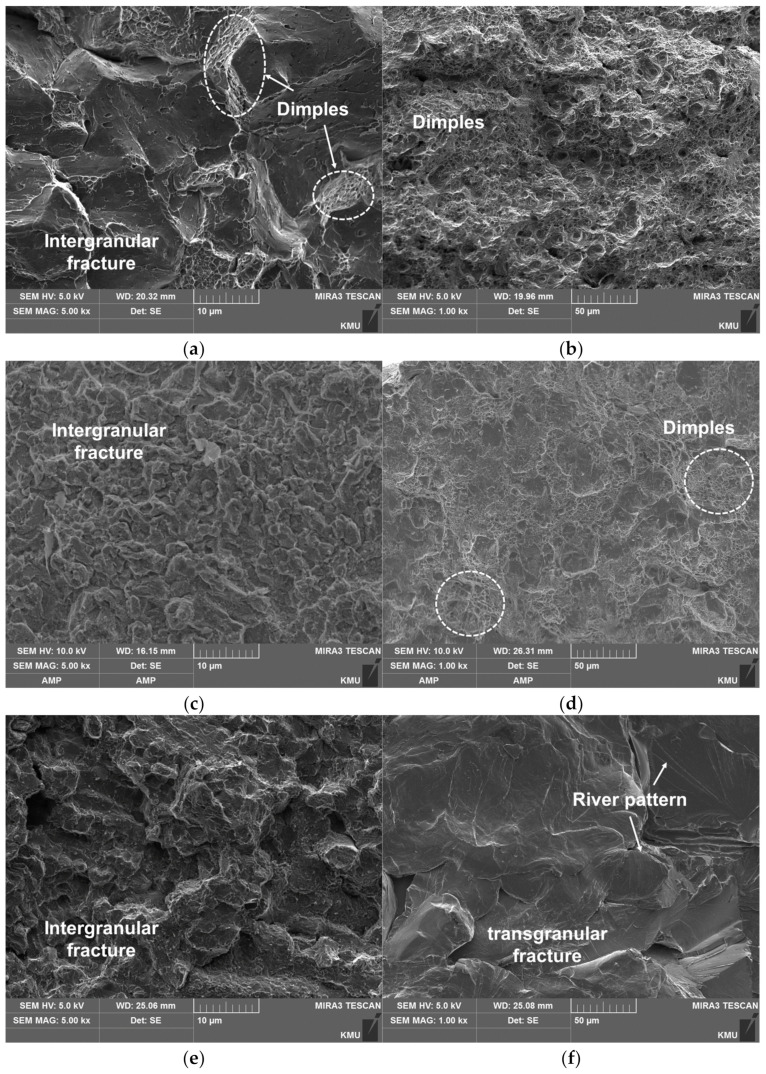
Fracture surfaces of the tensile specimens (**a**) carburized area of SCM420, (**b**) core of SCM420, (**c**) M2 deposited layer, (**d**) substrate of as-built (M2) specimen, (**e**) deposited layer of as-built (FeCrVMoW), (**f**) substrate of as-built (FeCrVMoW), (**g**) deposited layer of as-built (FeCrVMoW)-PHT and (**h**) substrate of as-built (FeCrVMoW)-PHT.

**Table 1 materials-14-01231-t001:** DED process conditions.

Laser Power (W)	Scanning Speed (mm/min)	Powder Feed Rate (g/min)	Coaxial Gas (L/min)	Shielding Gas (L/min)
800	850	5	8.0	2.5

**Table 2 materials-14-01231-t002:** Chemical composition of materials (wt%).

	C	Si	Mn	Mo	Cr	V	W	P	S	Ni	Cu
**Fe-8Cr-3V-2Mo-2W (Powder)**	1.08	1.38	0.34	1.86	7.80	2.66	1.73	-	-	-	-
**AISI M2 (Powder)**	0.803	0.16	0.29	4.84	3.98	0.90	5.84	0.018	0.013	0.07	-
**SCM420 (Substrate)**	0.19	0.23	0.80	0.17	1.00	-	-	-	-	0.02	0.02

## Data Availability

Data sharing is not applicable to this article.

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
