# Peer review of "Microstructures and Mechanical Properties of Deposited Fe-8Cr-3V-2Mo-2W on SCM420 Substrate Using Directed Energy Deposition and Effect of Post-Heat Treatment"

_materials, 2021, doi:10.3390/ma14051231_

Round 1
Reviewer 1 Report
The article "Microstructures and Mechanical Properties of Deposited Fe- 2
8Cr-3V-2Mo-2W on SCM420 Substrate using Directed Energy 3 Deposition and Effect of Post- Heat treatment" is an interesting article which focus on the mechan- 12 ical properties of the deposited Fe-8Cr-3V-2Mo-2W and the effect of heat treatment on it.
Several methods for the microsturulal and mechanical analysis used such as:
micro Vickers test
FE-SEM
Charpy
Tensile
XRD
EDS
Wear test
In my opinion, it would be better for the readers of the article, to modify the structure of the article.
Techniques for microstructure should be one subsection and techniques for mechanical properties should be another one.
Also, where they are missing, information on which equipment was used and what settings were used should be added to the materials section for each analysis method (example XRD - Bruker, Slits 1mm, 1mm, 1mm, Increment 0,01, step size 2, 2θ 5 - 90)
In XRD analysis Fig5, the 2θ axis must be the same in all samples.
Usually, we use a range between 5 - 90 degrees. At the other hand in Fig 5a the Internsity values are too low (maybe the slit was differents from the other samples) and the background in 5c and 5d is too high.
Reviewer 2 Report
This manuscript presents an efficient deposition technique accompanied by a post-heat process. It experimented with an attempt to reinforce the surface of the low-carbon-steel substrate with a new alloy powder. It disseminates the research work eloquently. Readers can get relevant information from this manuscript about the research on surface treatment.
The only concern about the conclusion is the authors claim the deposition technique described in this manuscript is giving similar or superior mechanical properties. Whereas this deposition technique is falling way behind in fracture toughness than the M2-deposited surface and the carburized SCM420. Please explain it.
The manuscript is both interesting and relevant to the research on materials and interesting to Material Science and Engineering community. I did not find any recent literature close to this topic. Hence, it seems original. Other published work continued using similar materials to reinforce the surface of the substrate. Whereas, this manuscript is attempting of using a new allow powder on a low-carbon steel substrate. Authors professionally addressed the main question posed.
Reviewer 3 Report
In this paper, the authors have investigated the microstructure and mechanical properties of Fe-8Cr-3V-2Mo-2W steel powder deposited through the directed energy deposition process on the SCM420 substrate.
As general remark, the paper is well organized.
1.Directed energy deposition (DED) is an additive manufacturing (AM) processes. In AM I recommend to use ISO / ASTM52900 – 15 Standard Terminology for Additive Manufacturing – General Principles – Terminology. You can cite:
International Organization for Standardization. ISO/ASTM 52900 Additive Manufacturing—General Principles—Terminology; International Organization for Standardization: Geneva, Switzerland, 2015.
Udroiu, R.; Braga, I.C.; Nedelcu, A. Evaluating the Quality Surface Performance of Additive Manufacturing Systems: Methodology and a Material Jetting Case Study. Materials 2019, 12, 995.
2.“chemical compositions of the substrate and Fe-8Cr-3V-2Mo-2W and M2 powders” Was this powder delivered by a supplier? If yes, please mention it. If no, how was determinate the powder composition?
3.Are the limitations of this study noted? The limitations of this study should be discussed.
4.Please mention the applications of this study.
Reviewer 4 Report
In this work, the Fe-8Cr-3V-2Mo-2W tool steel powder has been deposited on the SCM420 11 substrate through the DED process. Thereafter, the microstructure and mechanical properties of the deposited Fe-8Cr-3V-2Mo-2W before and after the heat treatment have been investigated. Afterwards, the influence of heat treatment on the hardness, wear, impact, and tensile tests have been studied and compared with the conventional one. The manuscript is well organized and well written. The design of the experiment, data collection, data analysis and discussion have been carried out adequately and so this manuscript can be accepted for publication in Materials. However, before accepting the following minor issues should be addressed:
- In the introduction, it is highly recommended to describe the DED process and it’s advantages in repair over the conventional repairing processes. For this reason, the following review paper can be used (https://doi.org/10.3390/app9163316).
- Which focus offset is used in this work?
- In Fig. 5(b) there are two peaks at the angles around 40 degrees that are not identified. What are those peaks for?
- The conclusion section is long and it should be shortened down including the key findings of the work.
